# The Influence of Droplet Distribution Coverage and Additives on the Heat Transfer Characteristics of Spray Cooling under the Influence of Different Parameters

Qian Niu [1], Yu Wang [1,]*[ID] and Na Kang [2]

1   College of Urban Construction, Nanjing Tech University, Nanjing 211816, China
2   College of Aerospace Engineering, Nanjing University of Aeronautics and Astronautics, Nanjing 210016, China
*   Correspondence: yu-wang@njtech.edu.cn; Tel.: +86-13913808658

**Abstract:** For the objective of enhancing the heat transfer ability of spray cooling, a single-nozzle open-loop spray cooling experiment platform was established with a data measuring system. Based on the surface heat transfer coefficient obtained from the experiment, combined with the visualization system to observe the distribution of droplets during the spray cooling process, the influence of heating power, medium flow rate, nozzle height and typical additives on heating surface coverage and heat transfer characteristics were investigated. The criterion non-dimensional criteria equations for Nu, Re, Pr and size coefficients were fitted and analyzed in comparison with experimental data. The main conclusions are as follows: considering the temperature distribution characteristics of the heating surface and the shape of the spray cone, the heat transfer performance can be optimized by increasing the coverage rate under high heat flux when the flow rate changes, appropriately reducing the coverage rate under low heat flux, and appropriately reducing the coverage rate when the height changes, which creates complete coverage in the droplet concentration area to improve the surface heat transfer capacity. Furthermore, the heat transfer coefficients were improved by 29.3%, 21.8% and 23.8% with different additives (CTAB, ethanol and CTAB–ethanol mixtures) in the working fluid. Each solution had an optimal concentration and heat transfer deterioration was observed at high concentrations. When using non-dimensional criteria equations for parameter calculations, the data are more accurate after considering the effect of dimensional coefficients.

**Keywords:** spray cooling; visualization system; heat transfer performance; surface coverage; additives



## 1. Introduction

With the increasing complexity, integration and miniaturization of airborne electronic equipment, the heat generation of electronic equipment is growing exponentially, and the impact of temperature on the normal operation of electronic equipment is indisputable. Faced with the challenge of high heat flux, the cooling requirements of airborne electronic devices have been difficult to satisfy using traditional cooling methods. The cooling capacity required to dissipate heat from airborne electronics is 50 kW, accounting for more than 85% of the total cooling capacity of the airborne temperature control system [1]. The normal operation of electronic equipment has high requirements for its own temperature and ambient temperature. Some investigations show that in the range of 70–80 °C, the reliability of electronic equipment will be reduced by 50% for every 10 °C increase in temperature [2]. The data show that IC (Integrated Circuit) failures related to heat dissipation account for more than 50% of all IC failures [3]. Since 2007, it has been demonstrated that many electronic devices have faced the challenge of maintaining a temperature below 85 °C at a heat flux of around 300 W/cm² [4]. The heat flux per unit volume of some high-power chips can exceed 500 W/cm² [5]. Application of a new and efficient heat transfer method is a matter of urgency. In the year 2017, the heat flux of high-performance electronic chips

was expected to reach 190 W/cm$^2$ by 2020 [6]. What is worse, the microminiaturization of the electronic equipment has greatly shrink the heat dissipation area, which has led to a heat flux of up to $10^3$ W/cm$^2$ in recent years [7].

Spray cooling is based on phase change heat transfer, which has the advantages of strong heat exchange capacity, low coolant flow, good homogeneity, low superheat and high critical heat flux (up to $10^3$ W/cm$^2$), as one of the most competitive technologies in the field of high heat flux cooling and airborne electronic equipment cooling. Closed-loop spray cooling technology is currently listed by NASA as one of the research priorities for future airborne thermal control systems [8], and the investigation has successfully approached the goal of $10^3$ W/cm$^2$ [9]. Spray cooling is widely used in medical, metallurgy, chip cooling, aerospace and other fields [10], and liquid nitrogen spray cooling can be used in some specific environments to achieve greater cooling capacity instantaneously [11]. The non-uniformity of droplet distribution in spray cooling and its effect on heat transfer performance were investigated by Cheng et al. [12] and Xie et al. [13]. Meanwhile, the effects of nozzle arrays [14], splash injection [15], and liquid film flow [16] were studied by some scholars. In each case, references are provided for the enhancement of spray heat transfer.

The study of the heat transfer flow mechanisms of spray cooling has been emphasized by researchers in the field of spray cooling in an effort to increase the heat transfer capacity. Using water as a medium for spray cooling trials, Liao et al. [17] concluded that the heat flux on the surface of the simulated heat source increased with increasing heating power. However, an air film between the heating surface and the liquid droplets prevented heat transfer. In a spray cooling experiment, Kim (2018) demonstrated that the critical heat flux increased with an increase in liquid flow rate, but the effect of liquid subcooling on critical heat flux was not significantly altered. Chen et al. [18] examined how micro-hole outlet diameter, volumetric flow rate, and spray height affected surface temperature distribution, heat flux, and spray cooling efficiency, and found that the heat flux increased as the flow rate increased, but the spray cooling efficiency decreased, and that there was an optimal orifice diameter and spray height for each nozzle. Cheng et al. [19] reviewed the progress of research on the heat transfer effect of spray cooling, and concluded that the variation in spray flow rate leads to the variation of other factors that result in the influence of heat transfer and need to be analyzed according to the experimental conditions. Both Mudawar et al. [20] and Visaria et al. [21] concluded that the highest heat transfer efficiency occurs when the spray impingement surface is directly parallel to the heating surface. However, by analyzing the surface structure of the heating surface and spray velocity characteristics through experiments and simulations, Cheng et al. [22] determined that a spray coverage of less than 100% would provide the maximum heat transfer efficiency. In view of the above studies, it has been determined that parameters such as spray cooling height and coverage have impacts on the heat transfer characteristics. It is unclear how droplets are distributed on the surface and influence the heat transfer parameters, and there has been some disagreement on the optimal coverage rate for the best heat transfer performance.

Some scholars worked on adding various additives, such as nanoparticles [23], alcohols [24], and surfactants [25], to water to improve the heat transfer efficiency of pure water. Johnathan [26] firstly performed pool boiling and spray cooling experiments using ethanol–water mixtures and measured contact angles. Ravikumar et al. [27] used Tween20 and ethanol as cooling media to study the cooling rate of air atomized spray on stainless steel plates with an initial temperature of 900 °C. It was shown that increasing the ethanol content in pure water and surfactant–water mixture can effectively increase the heat transfer coefficient and cooling rate. Liu et al. [28] investigated the heat transfer performance of AOS, CTAB and Tween20 in the low temperature range below 100 °C. The three surfactants had different optimal concentrations for heat transfer promotion, and heat transfer deterioration occurred at high concentrations due to bubble accumulation.

The use of tilt mode sprays can effectively alleviate bubble accumulation and thus improve heat transfer.

Zeitoun et al. [29,30] investigated the heat transfer between $Al_2O_3$ nanofluid jets and horizontal circular surfaces and showed that presenting the data in Reynolds number captures the effect of nozzle height and jet diameter, and presenting the data in Pecoret number correlates the nanofluid concentration with a single criterion number.

An important contribution of this paper is to examine the effect of different droplet distribution coverage on transient and steady-state heat transfer coefficients under a variety of parameter changes, including the effect of droplet coverage on heat transfer coefficients for medium flow rate changes and nozzle height changes. A combination of visualization analysis under the appropriate parameters and comprehensive consideration is used to determine the best spray coverage adjustment method for enhancing heat transfer. Based on the investigation of spray coverage rate, ethanol, water and the cationic surfactant hexadecyltrimethylammonium bromide (CTAB) were used as additives to investigate the enhancement effect of the above fluids and their mixtures on the heat transfer performance. Combined with the visualization analysis, the optimal concentrations of several additives to enhance the surface heat transfer were synthesized.

## 2. Materials and Methods

### 2.1. Experimental Equipment

An experimental system for spray cooling is described, including a spray medium system, a simulated heating system, a visualization system, and a data acquisition system. The schematic diagram of the system is shown in Figure 1.

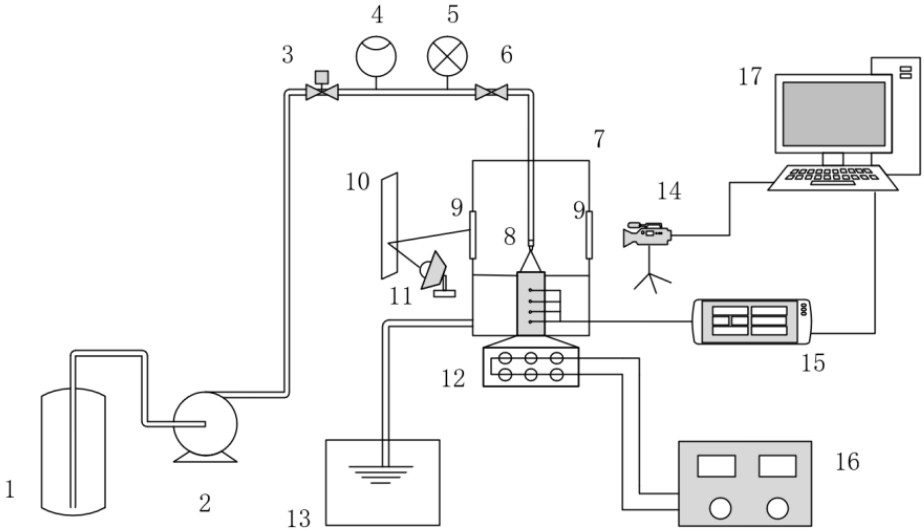

**Figure 1.** Schematic diagram of spray system. 1 liquid storage tank; 2 micro high-pressure pump; 3 flow adjustment valve; 4 flow meter; 5 pressure gauge; 6 stop valve; 7 spray chamber; 8 nozzle; 9 observation window; 10 reflector; 11 fill light; 12 heating rod; 13 waste liquid collection tank; 14 high-speed camera; 15 data acquisition instrument; 16 heating power regulator; 17 computer.

As shown in Figure 1, the cooling medium in the reservoir is pumped by a micro high-pressure pump, flows through the flow adjustment valve, flow meter, and is sprayed out through the nozzle to the surface of the simulated heat source. The cooling medium after heat exchange flows into the waste liquid collection tank after treatment. The high-speed camera is fixed by the bracket in front of the observation window. The fill light and reflector are installed in front of the observation window on the other side for illumination. Using a diffuse reflector, heat emitted by light coming into the chamber can be reduced to prevent temperature changes on the surface of the heat source. A certain mass or volume of solute

is placed in a quantitative amount of pure water and dissolved with stirring to obtain a solution of specified concentration.

The simulated heating block is constructed from purple copper, and the upper heat source surface consists of a circle with a diameter of 2.4 cm. The temperature and heat flux on the surface of the simulated heat source are changed by adjusting the heating power or heat flux through a heating power regulator. The maximum heating power of the simulated heat source is 1800 W, and the maximum heat flux is 398 W/cm$^2$. The flow rate of the medium can be changed by changing the flow adjustment valve and observing the flow meter, and the heat on the heating surface is exchanged with the spray droplets. Four K-type thermocouples with 8 mm spacing are arranged vertically in the heating block for temperature measurement, as shown in Figure 2, and the temperature is collected by a Keysight DAQ970A data acquisition instrument (Keysight Technologies, Inc., Santa Rosa, CA, USA). During the experiment, the internal image of the spray chamber is captured in real-time by the OSG030-815UM (Weihong Image (Shenzhen) Co., Ltd., Shenzhen, China) industrial high-speed camera of the visualization system.

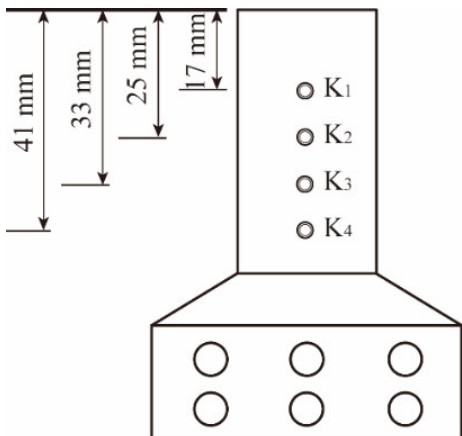

**Figure 2.** Schematic diagram of the vertical arrangement of thermocouples.

*2.2. Data Processing*

Aluminum silicate fiber wool is installed around the heating block to isolate the heating block from the external thermal environment, so the heat transfer from the side of the heating block can be ignored and the temperature change inside the heating block follows the one-dimensional thermal conductivity law. The heat flux at the surface of the simulated heat source can be obtained from Fourier's one-dimensional thermal conductivity law as shown in Equation (1).

$$q = -\lambda \frac{\Delta T}{\Delta y}, \tag{1}$$

where $q$ is the heat flux of the simulated heat source surface, W/cm$^2$, $\lambda$ is the thermal conductivity of copper, W/(cm·K), and $\frac{\Delta T}{\Delta y}$ is the temperature distribution gradient in the axial direction of the heating block.

Linear fitting of the temperature data from the four thermocouples yields the Equation (2)

$$T(y) = a + by, \tag{2}$$

The surface temperature of the heat source as shown in Equation (3).

$$T_W = T_1 + by_1, \tag{3}$$

where $T_W$ is the temperature at the surface of the simulated heat source, °C, $T_1$ is the temperature measured by thermocouple $K_1$, °C, and $y_1$ is the distance between thermocouple $K_1$ and the surface of the heat source, mm.

Figure 3 shows the linear fit analysis of one set of experimental data, and it can be seen that the data points are in good agreement with the fitted line, so the fitting results are reliable.

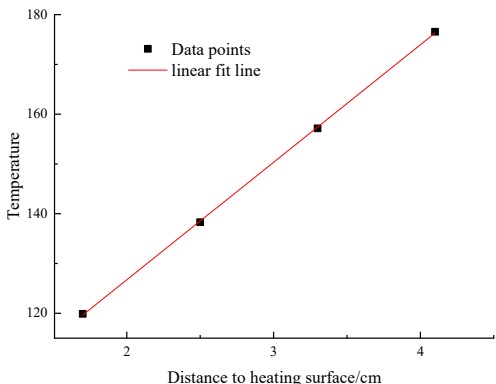

**Figure 3.** Schematic of the linear fit of the longitudinal temperature.

The heat transfer coefficient on the simulated surface of the heat source is shown in Equation (4)

$$h = \frac{q}{T_W - T_{in}}, \tag{4}$$

where $h$ is the surface heat transfer coefficient, W/(cm$^2$·K), $T_{in}$ is the nozzle inlet temperature of the cooling medium, °C.

A criterion-dependent correlation fit is performed at the spray impact point, and the temperature of the calculated fluid parameters depends on the average of the heated surface temperature and the medium temperature:

$$T = \frac{T_W + T_{in}}{2}, \tag{5}$$

The thermal conductivity of the mixed fluid is

$$\lambda_m = \varphi_1 \lambda_1 + \varphi_2 \lambda_2 - (\lambda_2 - \lambda_1)(1 - \varphi_2^{0.5})\varphi_2, \tag{6}$$

where $\varphi_i$ is the mass concentration of each component, $\lambda_i$ is the thermal conductivity of each component, W/(m·K).

The dynamic viscosity of the mixed fluid is

$$\mu_m = \left(\sum_{i=1}^{n} \varphi_i \mu_i^{\frac{1}{3}}\right)^3, \tag{7}$$

where $\mu_i$ is the dynamic viscosity of each component, Pa·s.

The density of the mixed fluid is

$$\rho_m = \rho_1 \varphi_1 + \rho_2 \varphi_2, \tag{8}$$

where $\rho_i$ is the density of each component, kg/m$^3$.

The specific heat at constant pressure of the mixed fluid is

$$c_{pm} = \frac{c_{p1} \varphi_1 \rho_1 + c_{p2} \varphi_2 \rho_2}{\rho_m}, \tag{9}$$

where $Cp_i$ is the constant pressure specific heat of each component, J/kg·K.

Re is calculated using the equation related to mass flux:

$$\text{Re} = \frac{GD}{\mu}, \tag{10}$$



where $D$ is the spray coverage diameter, m, $\mu$ is the medium dynamic viscosity, Pa·s, $G$ is the mass flux of cooling medium, kg/(s·m$^2$).

The mass flux $G$ is calculated by the equation:

$$G = \frac{G_m}{A},\tag{11}$$

where $G_m$ is the mass flow rate of the medium, kg/s, $A$ is the area covered by the spray, m$^2$.

Nusselt number is:

$$Nu = \frac{hD}{\lambda},\tag{12}$$

where $h$ is the surface heat transfer coefficient, W/(m$^2$·K), $D$ is the spray coverage diameter, m, $\lambda$ is the thermal conductivity, W/(m·K).

Prandtl number is:

$$\Pr = \frac{\mu c_p}{\lambda},\tag{13}$$

where $\mu$ is the medium dynamic viscosity, Pa·s, $Cp_i$ is the constant pressure specific heat of each component, J/(kg·K).

### 2.3. Uncertainty Analysis

K-type thermocouple is used to measure the temperature of 4 measurement points on the heating block with an uncertainty of ±0.8 °C. The uncertainty of the temperature gradient of fitting 4 thermocouples is ±0.01 °C/mm. PT100 platinum resistance is used to measure the temperature of the spray outlet mass with an uncertainty of ±0.15 °C. Due to the limitation of the laser drilling process, the uncertainty of the thermocouple mounting position is ±0.1 mm.

According to the error transfer equation proposed by Kline et al. (1953). The error analysis is shown in Equation (14).

$$\frac{\delta R}{R} = \sqrt{\left(\frac{\delta X_1}{X_1}\right)^2 + \left(\frac{\delta X_2}{X_2}\right)^2 + \cdots + \left(\frac{\delta X_M}{X_M}\right)^2},\tag{14}$$

where $R$ is the parameter for which the error needs to be calculated, and $X_1{\sim}X_M$ are all the variables associated with this parameter.

The uncertainty of the heat flux can be known as Equation (15):

$$\frac{\delta q}{q} = \sqrt{\left(\frac{\delta\lambda}{\lambda}\right)^2 + \left(\frac{\delta T}{T}\right)^2 + \left(\frac{\delta y}{y}\right)^2 + \left(\frac{\delta b}{b}\right)^2},\tag{15}$$

where $q$ is the heat flux of the simulated heat source surface, W/cm$^2$, $\lambda$ is the thermal conductivity of copper, W/(cm·K), $T$ is the measured temperature of the thermocouple, °C, $y$ is the distance of the thermocouple from the surface of the heat source, mm, and $b$ is the slope value after linear fitting of the temperature, °C/mm.

The uncertainty of the simulated heat source surface temperature is shown in Equation (16):

$$\frac{\delta T_W}{T_W} = \sqrt{\left(\frac{\delta T_1}{T_1}\right)^2 + \left(\frac{\delta\Delta T}{\Delta T}\right)^2},\tag{16}$$

where $T_W$ is the temperature at the surface of the simulated heat source, °C, $T_1$ is the temperature measured by thermocouple K$_1$, °C, and $\Delta T$ is the distance between thermocouple K$_1$ and the surface of the heat source, mm.

The uncertainty of the surface heat transfer coefficient is shown in Equation (17):

$$\frac{\delta h}{h} = \sqrt{\left(\frac{\delta q}{q}\right)^2 + \left(\frac{\delta T_W}{T_W - T_{in}}\right)^2 + \left(\frac{\delta T_{in}}{T_W - T_{in}}\right)^2},\tag{17}$$

where $h$ is the surface heat transfer coefficient, W/(cm$^2$·K), $q$ is the heat flux of the simulated heat source surface, W/cm$^2$, $T_W$ is the temperature at the surface of the simulated heat source, °C, $T_{in}$ is the nozzle inlet temperature of the cooling medium, °C.

The uncertainties of the experimental heat flux, surface temperature and surface heat transfer coefficient were calculated to be 3.8%, 2.7% and 5.2%, respectively.

## 3. Results and Discussion

### 3.1. Effect of Heat Flux Variation on Heat Transfer Performance

The effects of different heat fluxes of the simulated heat source on the surface heat transfer coefficient at a flow rate of 40 L/h for a nozzle with an orifice diameter of 1 mm are investigated in this section.

As shown in Figure 4, the transient surface heat transfer coefficient increases with the increase in the heating heat flux during the overall heating process. The surface heat transfer coefficient increases monotonically with time at a heat flux of 50 W/cm$^2$ and stabilizes at 50 min. In the process of changing the transient surface heat transfer coefficient with heat flux of 100 W/cm$^2$, 150 W/cm$^2$ and 200 W/cm$^2$, it appears that the heat transfer coefficient first rises and then decreases, and finally stabilizes. The trend is more obvious in the case of 150 W/cm$^2$ and above. Combined with the visualization and analysis process, it can be found that when the system starts to operate, the surface heat flux increases sharply, causing a significant increase in the heat transfer coefficient. With increasing heat flux, smaller droplets evaporate rapidly on the heating surface to produce a vapor layer, and the heat exchange between some droplets and the heat source surface is resisted by the vapor. As time increases, as the droplet's own velocity overcomes the effect of vapor on heat exchange, the hindering effect of a small number of small droplets evaporating and the overall cooling heat exchange reach a state of equilibrium. Therefore, the transient heat transfer coefficients tend to be stable after 60 min.

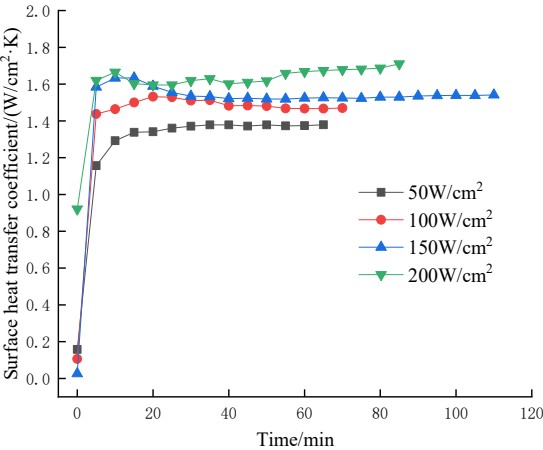

**Figure 4.** Transient surface heat transfer coefficient with change of heat flux at a flow rate of 40 L/h.

The effect of heat flux variation on the steady-state surface heat transfer coefficient is shown in Figure 5. The steady-state surface heat transfer coefficient increases with the increase in the heating heat flux. The simulated heat source surface heat flux is directly dependent on the heating heat flux, Therefore, when the heating heat flux is small, the surface temperature and heat flux of the simulated heat source are small, and the heat exchange between the heat source surface and the spray droplets is carried out in the non-boiling zone with a small heat transfer coefficient. With the increase in the heating heat flux, the simulated heat source surface heat exchange enters the boiling zone, and the larger the surface heat flux, the greater the boiling degree, and the heat transfer coefficient is the greater.

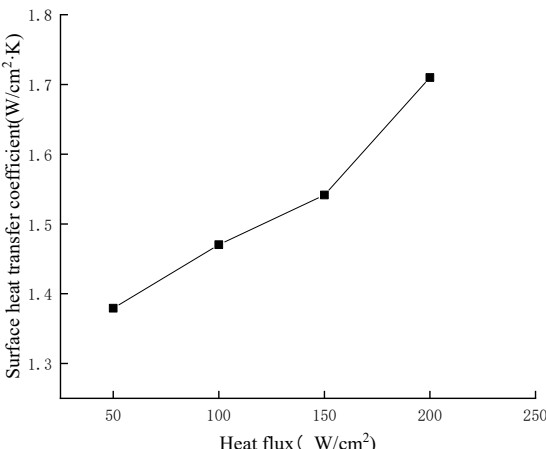

**Figure 5.** The influence of the change of heating heat flux on the steady-state surface heat transfer coefficient at a flow rate of 40 L/h.

The visualization images of the spray chamber at heating heat fluxes of 50 W/cm², 100 W/cm², 150 W/cm², and 200 W/cm² are shown in Figure 6. It can be observed that there is little difference in the internal state of the spray chamber at low heat fluxes. In the heating heat flux of 150 W/cm², the surface temperature of the heat source has reached the boiling point of water, and water vapor appears in the chamber. The entire chamber is filled with water vapor when the heating heat flux is 200 W/cm². The increase in water vapor causes the pressure inside the chamber to increase, and the boiling point of water to rise, which to a certain extent blocks the boiling heat exchange on the surface of the heat source. Therefore, fluctuations in the heat transfer coefficient can occur when heating with large heat flux.

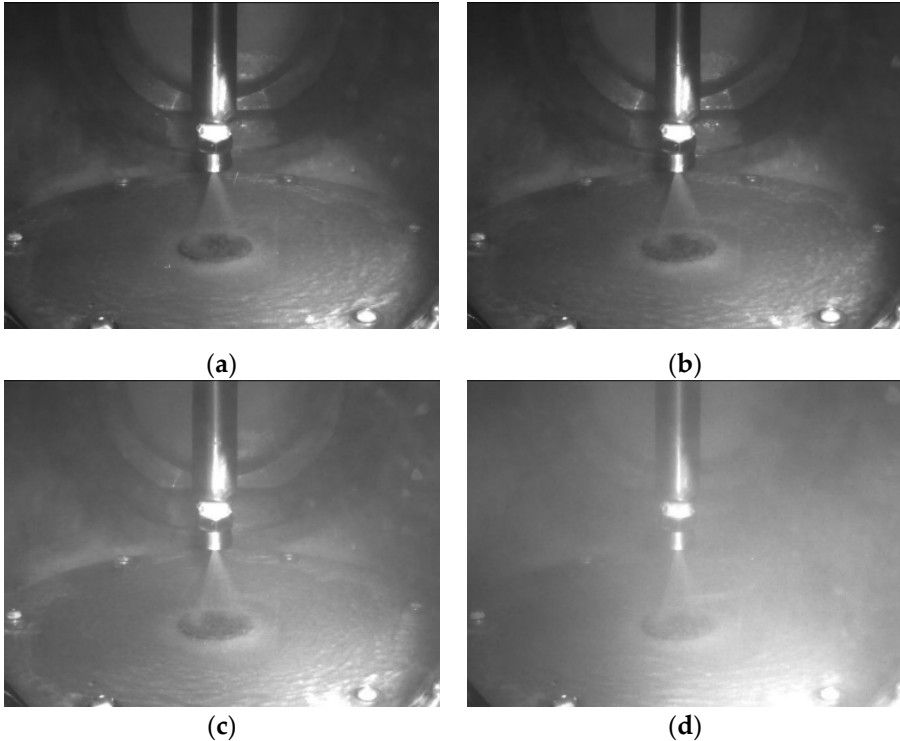

**Figure 6.** Visualization images of the influence of the change of heating heat flux on the inside of the spray chamber at a flow rate of 40 L/h: (**a**) 50 W/cm²; (**b**) 100 W/cm²; (**c**) 150 W/cm²; (**d**) 200 W/cm².

### 3.2. Effect of Coverage on Heat Transfer Performance by Changing Flow Rate

The visualization images of the interior of the chamber with the flow rate of 25 L/h, 30 L/h, 35 L/h and 40 L/h with 100 W/cm$^2$ heating heat flux are shown in Figure 7. As can be seen from the nozzle product manual and computer measurements, in this experimental system, when the Ge Qiang 1/8 1 mm nozzle flow rate is 25 L/h, the spray cone angle is 25°, as illustrated in Figure 7a. The droplet jet cannot overcome gravity's acceleration when the flow rate is too low, so a complete spray cone cannot be formed and a corresponding coverage cannot be calculated. The spray cone angle is 30° when the flow rate is 30 L/h, 40 degrees when the flow rate is 35 L/h, and 45 degrees when the flow rate is 40 L/h. By setting nozzle coverage to 100% at a flow rate of 40 L/h, it is possible to determine a nozzle height of 2.89 cm. With this height fixed, reducing the flow rate to 35 L/h and 30 L/h, the corresponding coverage rates can be calculated as 77% and 42%, respectively.

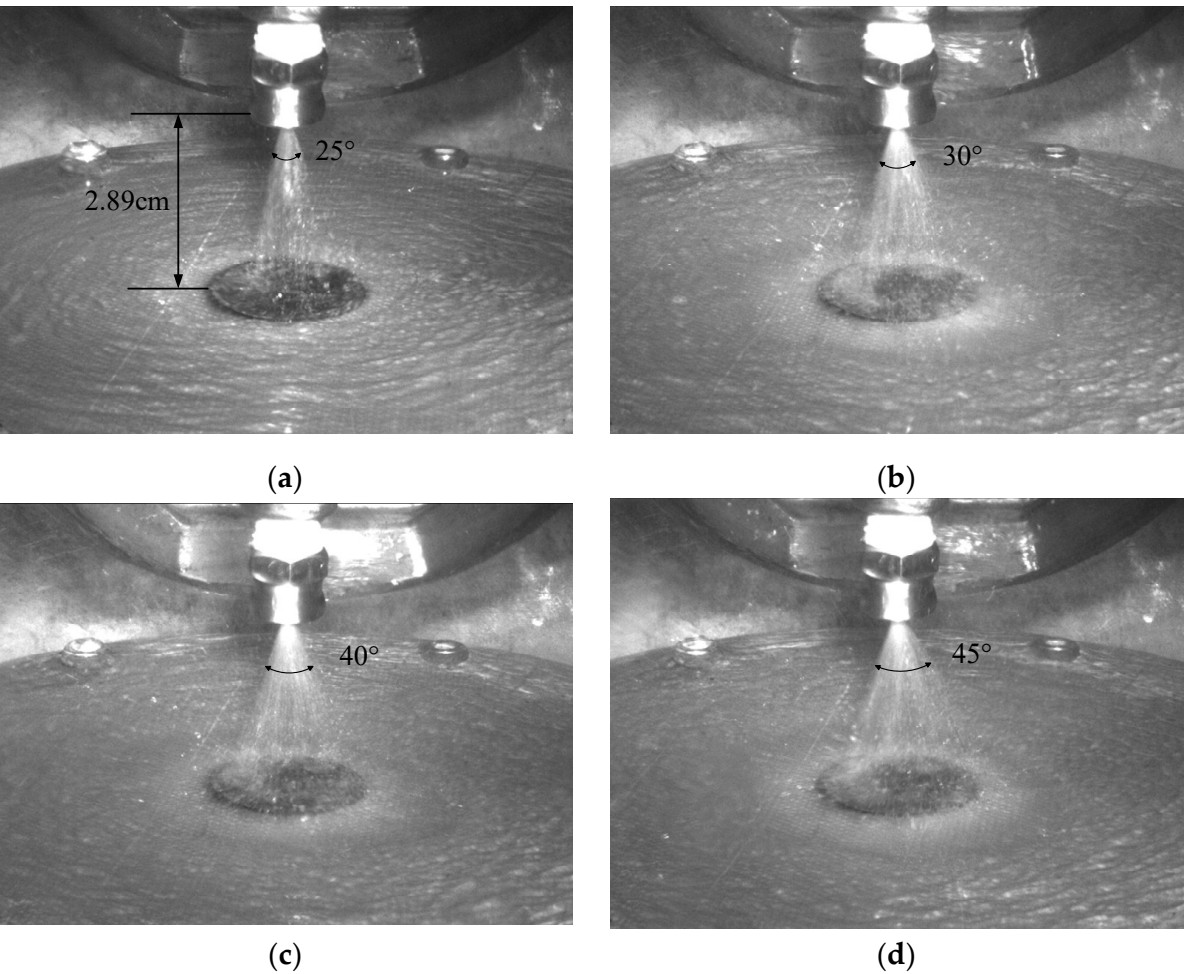

(a)　　　　　　　　　　　　　　　　　　(b)

(c)　　　　　　　　　　　　　　　　　　(d)

**Figure 7.** Visualization images of the influence of flow rate change on spray coverage at a heating heat flux of 100 W/cm$^2$: (**a**) flow rate 25 L/h; (**b**) flow rate 30 L/h, coverage 42%; (**c**) flow rate 35 L/h, coverage 77%; (**d**) flow rate 45 L/h, coverage 100%.

Figures 8–11 show the effect of coverage on transient surface heat transfer coefficient by changing flow rate at different heating heat fluxes (50–200 W/cm$^2$). From Figures 8–11, it can be seen that the transient surface heat transfer coefficient increases with the increase in coverage at different heating heat fluxes when the medium flow rate is increased, and the phenomenon is more obvious especially at high heat fluxes. When the heating heat flux is low, the heat exchange on the surface of the heat source is not completely in the boiling zone, the greater the flow rate, the greater the droplet velocity, and the greater the number of instantaneous droplets. As a result, at this time, some droplets that are not ready to

boil will prevent the surface from boiling, causing a fluctuation of the surface heat transfer coefficient at a higher flow rate. While when the heat flux is 150 W/cm² and above, the surface heat exchange is all carried out in the boiling zone, more droplets at higher flow rates will accelerate heat transfer. Hence this part of the surface heat transfer coefficient increases more smoothly, and the higher the flow rate, the larger the transient surface heat transfer coefficient. In addition, Figures 10 and 11 also verify the conclusion in Section 2.1 that the surface heat transfer coefficient first increases and then decreases and eventually plateaus at high heat fluxes.

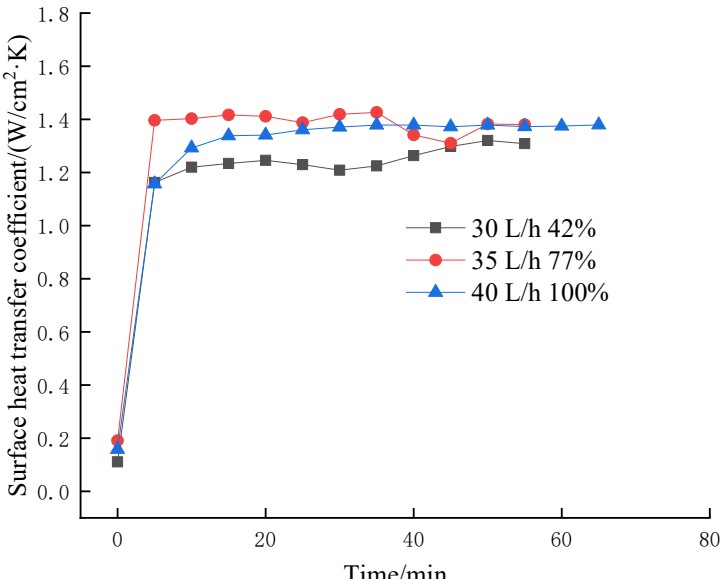

**Figure 8.** The influence of coverage rate of changing flow rate at 50 W/cm² on transient surface heat transfer coefficient.

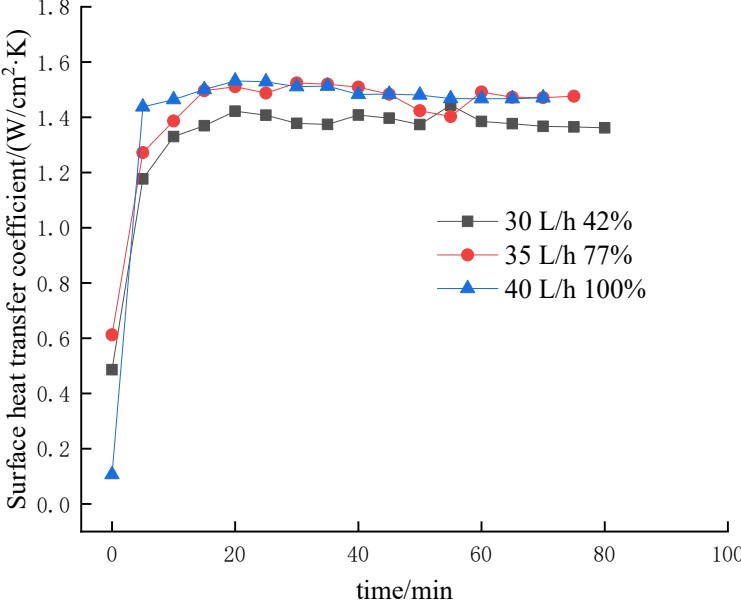

**Figure 9.** The influence of coverage rate of changing flow rate at 100 W/cm² on transient surface heat transfer coefficient.

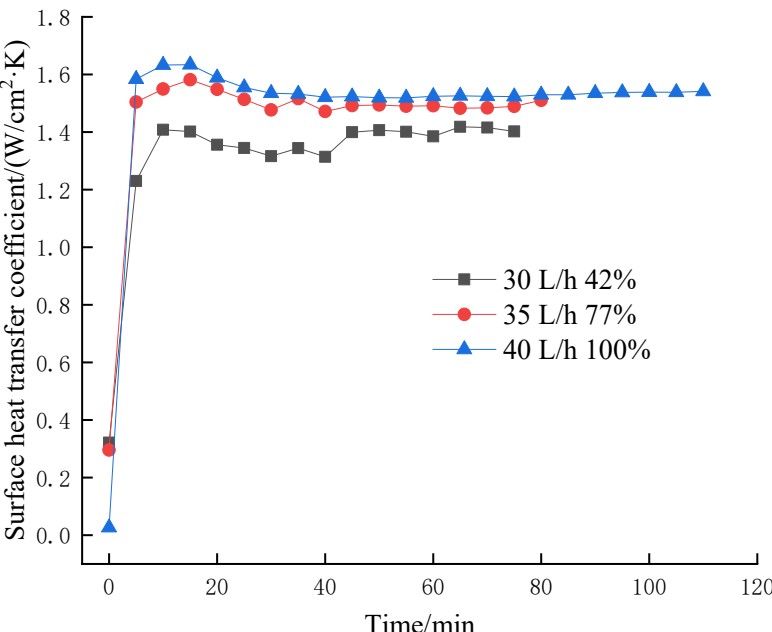

**Figure 10.** The influence of coverage rate of changing flow rate at 150 W/cm² on transient surface heat transfer coefficient.

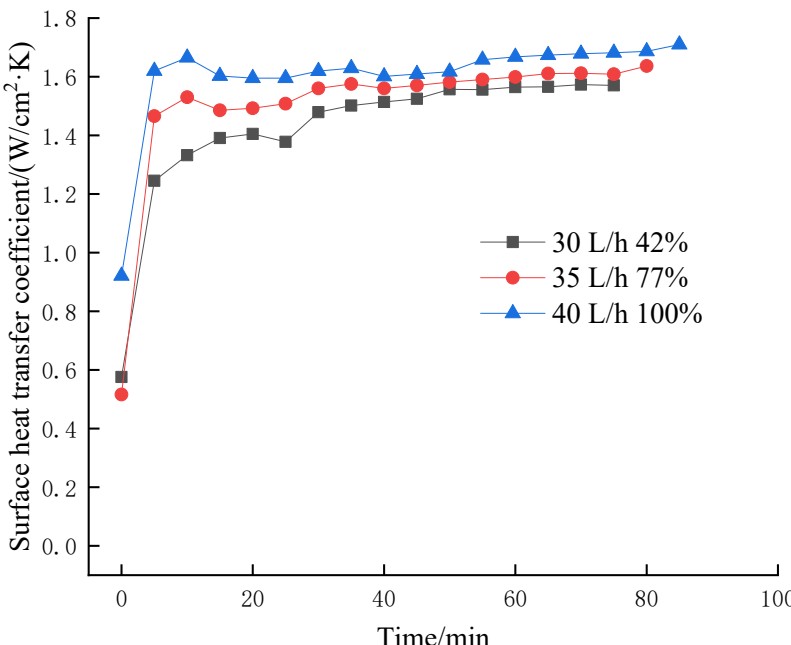

**Figure 11.** The influence of coverage rate of changing flow rate at 200 W/cm² on transient surface heat transfer coefficient.

Figure 12 shows the effect of coverage on steady-state surface heat transfer coefficient by changing flow rate effect of varying coverage by flow rate. It can be seen from the figure that the steady-state surface heat transfer coefficient increases with increasing coverage at higher heat fluxes. This coverage increases as the flow rate increases, so the greater the coverage, the greater the number of droplets on the heat transfer surface, the stronger the heat transfer effect in the boiling zone at high heat fluxes, and therefore the heat transfer coefficient increases. At the heat flux of 100 W/cm² and below, the surface heat transfer coefficient first increases and then slightly decreases with the increase in coverage. As can be seen from Figures 8–11, the transient surface heat transfer coefficient fluctuates for large

flow rates and small heating heat fluxes. Therefore, the steady-state heat transfer coefficient also decreases. Figure 12 also further verifies the above conclusion.

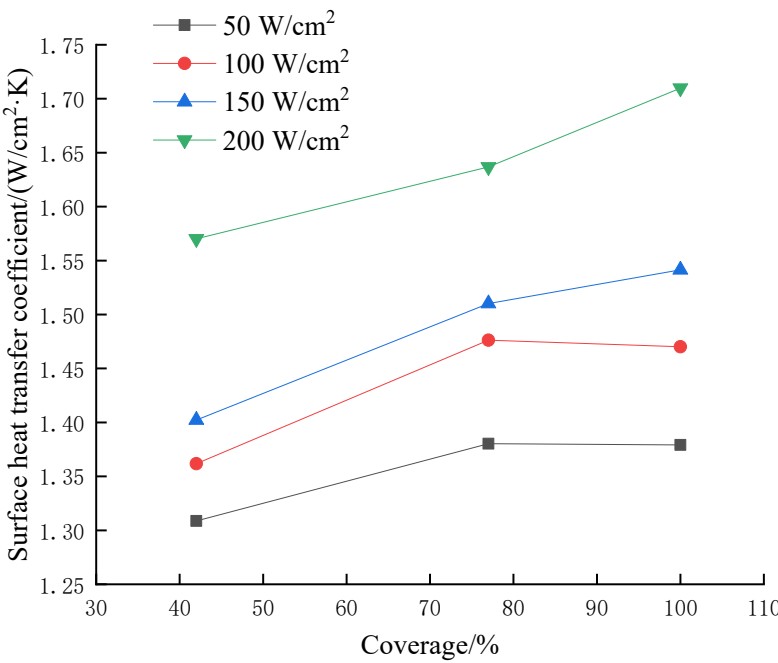

**Figure 12.** The influence of the coverage rate of the flow rate change on the steady-state surface heat transfer coefficient.

### 3.3. Effect of Coverage on Heat Transfer Performance by Changing Height

The 40 L/h flow rate is selected for the experiment, and 100% coverage is achieved at a height of 2.89 cm. Reducing the height by 20% in turn, the coverage corresponding to heights of 2.31 cm, 1.73 cm, and 1.16 cm are obtained as 64%, 36%, and 16%, respectively, as shown in Figure 13.

Figures 14 and 15 show the effect of coverage on transient surface heat transfer coefficient by changing height at different heating heat fluxes. It is obvious from the Figures that the heat transfer coefficient does not increase with the larger coverage, and it often appears that the heat transfer coefficient increases and then decreases with the increase in coverage.

The surface of the heating block is usually considered to have a uniform temperature distribution, but in fact the surface temperature of the heater decreases from the center to the edge along the radial direction, with the highest temperature in the center of the simulated heat source. Therefore, distributing the droplets evenly on the heating surface cannot achieve the best heat transfer effect. The nozzle spray droplets in the same section of the radial velocity is also different, the overall cross-sectional velocity curve will show a saddle surface [22], thus there is no uniform heat exchange over the entire heating surface. The droplets in the spray cone are not uniformly distributed, either. The droplets are dense and more uniformly distributed in the upper part of the cone, and in the middle and lower part of the cone, due to the entrainment effect arising from the inconsistent velocity between the center and the edge of the cone, there will be a parabolic development of the cone below, as shown in Figure 16. At this time, there are fewer droplets in the center of the cone, and the heating surface is not uniform in heat transfer.

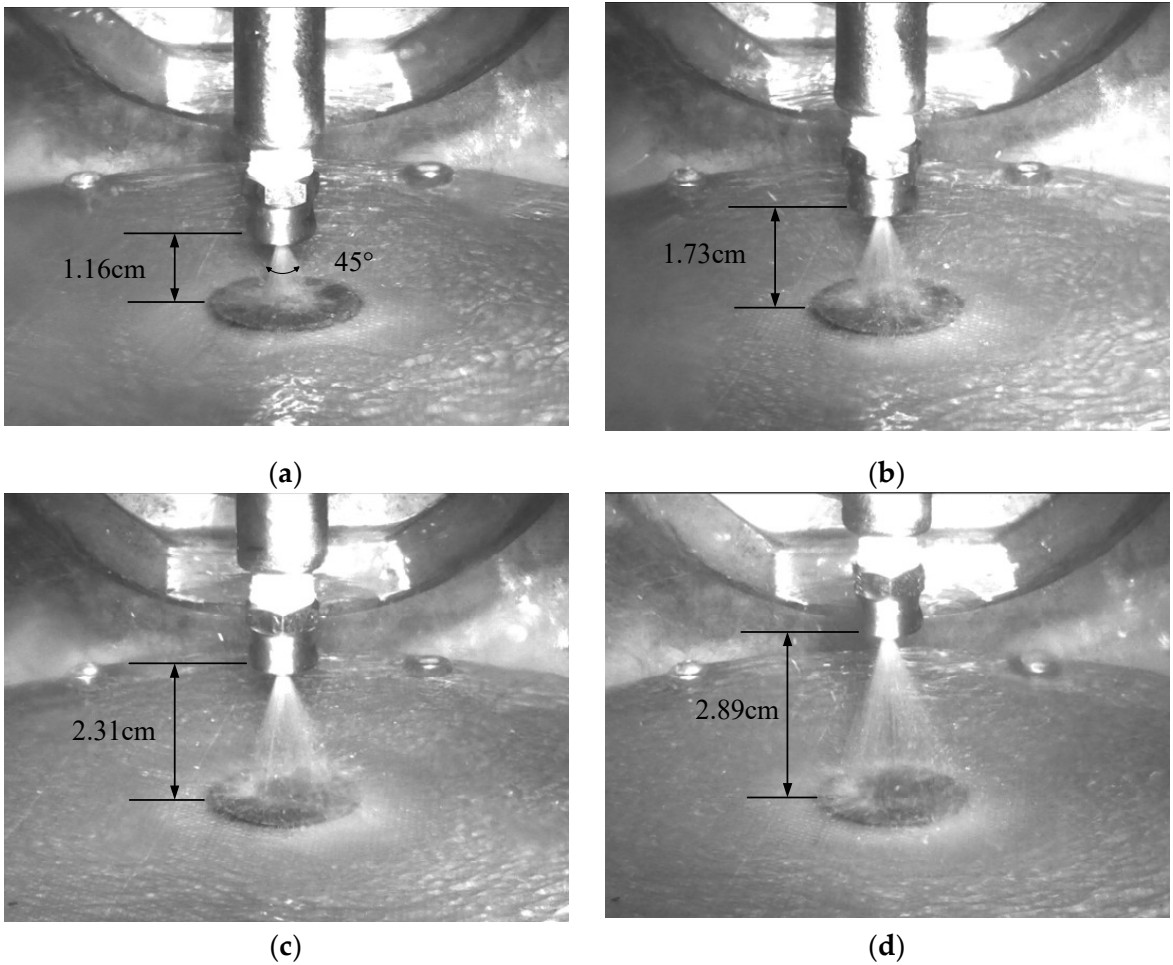

**Figure 13.** Visualization images of the effect of nozzle height change on spray coverage at a flow rate of 40 L/h: (**a**) height 1.16 cm, coverage16%; (**b**) height 1.73 cm, coverage 36%; (**c**) height 2.31 cm, coverage 64%; (**d**) height 2.89 cm, coverage 100%.

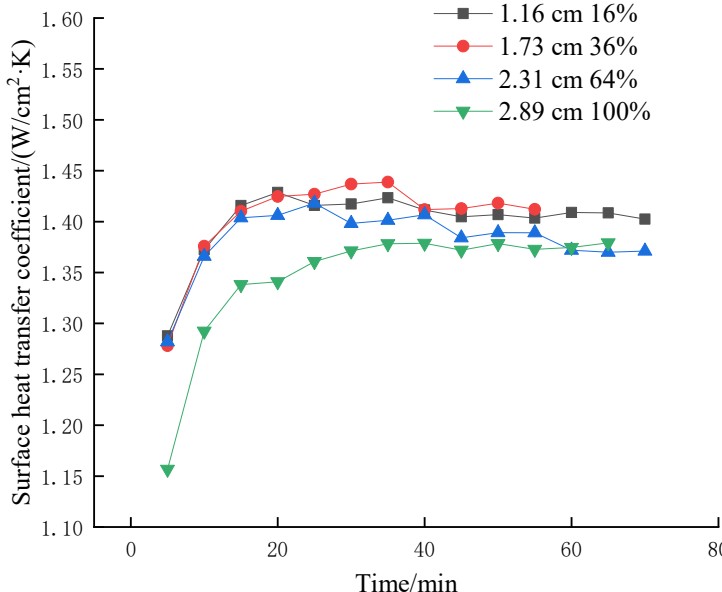

**Figure 14.** The influence of the coverage rate of changing height at 50 W/cm$^2$ on the transient surface heat transfer coefficient.

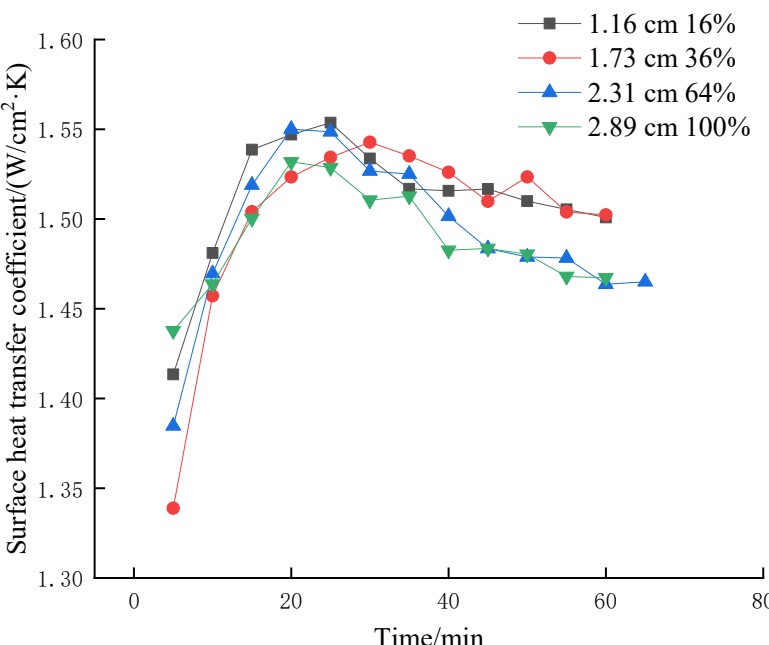

**Figure 15.** The influence of the coverage rate of changing height at 100 W/cm$^2$ on the transient surface heat transfer coefficient.

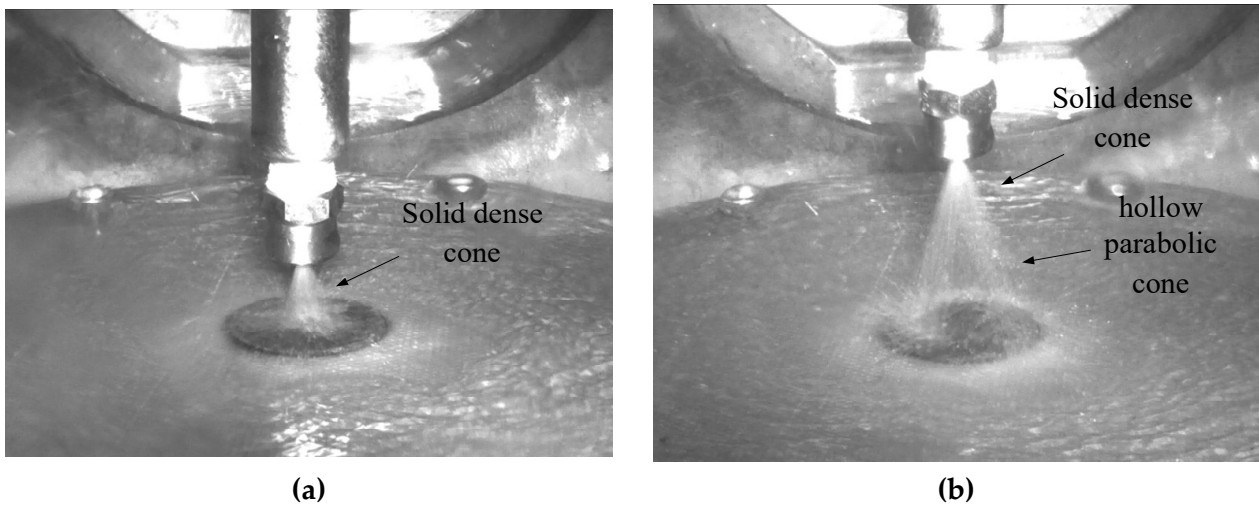

**Figure 16.** Visualization images of spray cone shape: (**a**) solid dense cone, coverage 16%; (**b**) solid dense cone and hollow parabolic cone.

Figure 17 shows the effect of coverage on steady-state surface heat transfer coefficient by changing height at different heating heat fluxes. As can be seen from the figure, the steady-state heat transfer coefficient reaches a maximum at a nozzle height of 1.73 cm and a coverage of 36%, after which the heat transfer coefficient curve decreases and tends to be steady. The heat transfer coefficient at 16% coverage is also greater than that at 64% and 100% coverage. The trend of the images at 50 W/cm$^2$ and 100 W/cm$^2$ is consistent. When the nozzle height is small, the upper part of the spray cone where the droplets are dense exchange heat with the heating surface at the center of the larger heat flux. The droplets in the center of the heating surface push the liquid to flow around to cover the whole heating surface, with good boiling state and high heat exchange efficiency. When the nozzle height is slightly increased, the heating surface remains in the dense droplet area and the droplet coverage area expands, thus the heat transfer coefficient increases. The nozzle height continues to be increased and the droplet coverage is increased, but the spray

cone above the heated surface has been parabolic, with fewer droplets in the center of the heated surface. At this time, better heat transfer performance only appears at the edge of the heating block. Thus, the overall heat transfer coefficient is reduced.

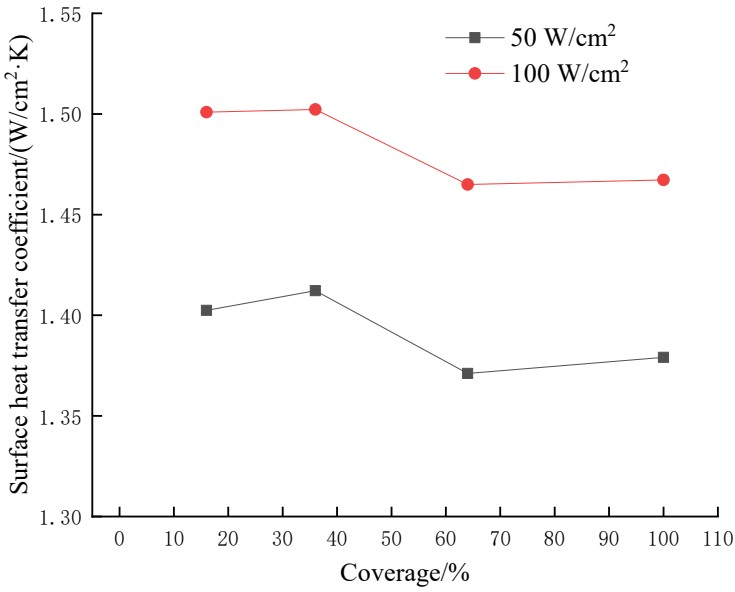

**Figure 17.** The influence of the coverage rate of the nozzle height change under different heat fluxes on the steady-state surface heat transfer coefficient.

### 3.4. Effect of Cationic Surfactant (CTAB)–Water Mixture on the Heat Transfer Performance

Figure 18 shows the steady-state heat transfer coefficients for spray cooling at different CTAB concentrations. It can be seen from Figure 18 that the surface heat transfer coefficient increases and then decreases with increasing CTAB concentration. The maximum heat transfer coefficient of 1.9 W/cm$^2$·K is reached at the concentration of 200 ppm, which is 29.3% higher than the heat transfer coefficient of 1.47 W/cm$^2$·K of pure water. Beyond this point, the concentration continued to increase, and the heat transfer coefficient decreased continuously, and when the concentration reaches 300 ppm, the heat transfer deteriorated.

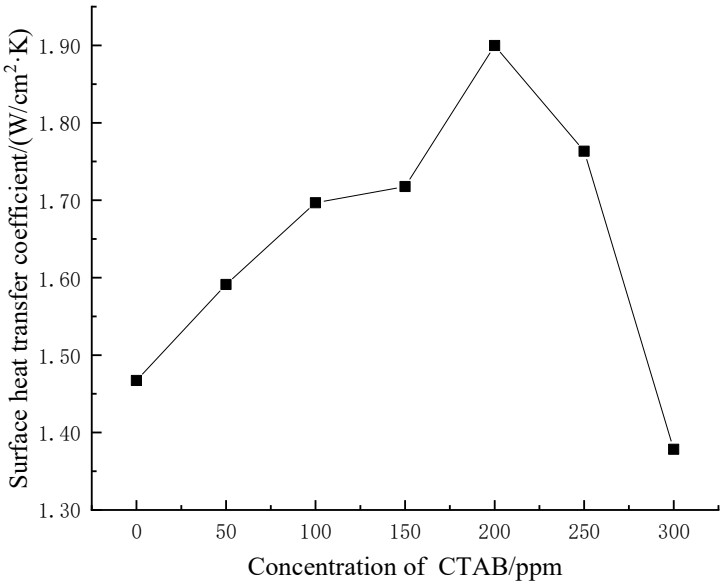

**Figure 18.** The steady-state heat transfer coefficients of spray cooling at different CTAB concentrations.

The phenomenon can be analyzed in conjunction with the visualization images of the chamber interior in Figure 19. The addition of surfactant reduces the surface tension and contact angle. The higher the concentration, the more the surface tension and contact angle are reduced. The reduction in surface tension makes it easier for the droplets to overcome the surface tension and break into smaller droplets. The increased droplet number is more conducive to driving the spreading of the liquid film on the heated surface. The decrease in contact angle increases the contact area between the droplets and the surface, increasing the evaporation rate of the liquid. The increase in droplet number density leads to an increase in the number of bubbles carried by the spray. The reduction in surface tension and contact angle at lower additive concentrations plays a dominant role in heat transfer. As a result, the heat transfer coefficient increases substantially.

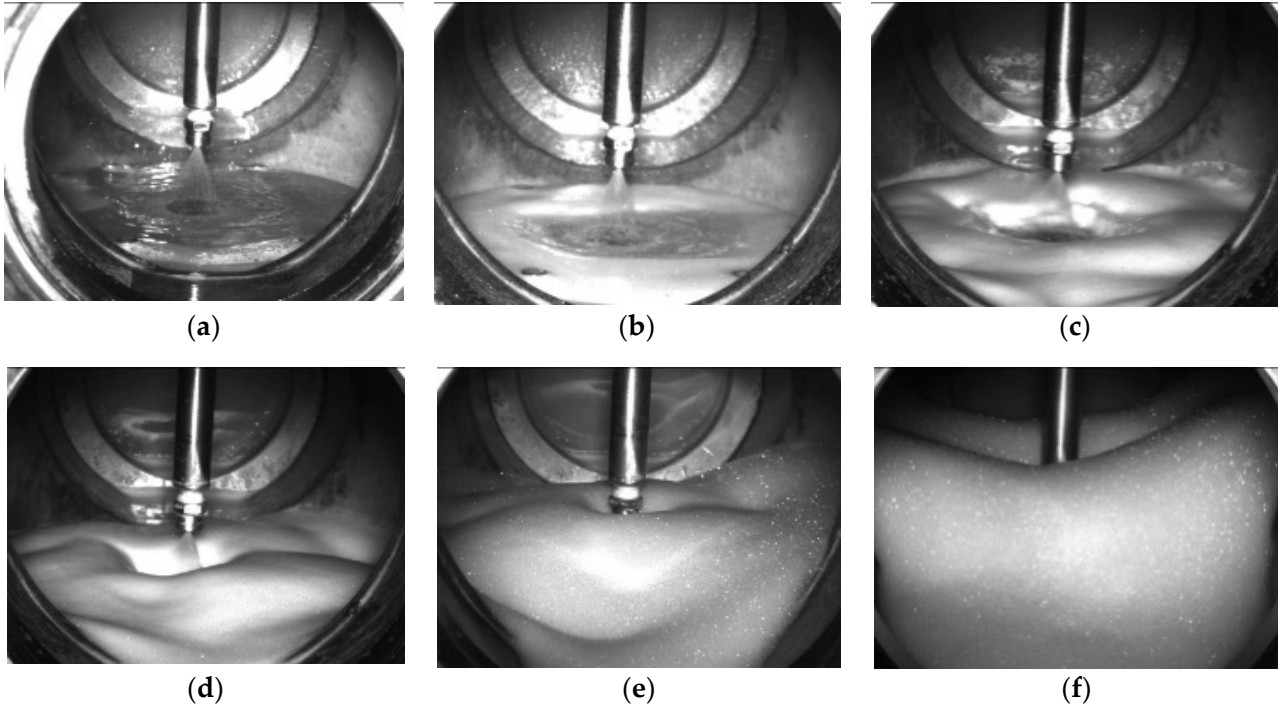

**Figure 19.** Visualization images of the foam inside the chamber at different CTAB concentrations: (**a**) 50 ppm; (**b**) 100 ppm; (**c**) 150 ppm; (**d**) 200 ppm; (**e**) 250 ppm; (**f**) 300 ppm.

After the concentration reaches the optimum value, the high foamability of the surfactant gradually dominate as the concentration continues to increase. As shown in Figure 19, the foaming inside the cavity gradually increases and hinders the flow of the surface liquid film. Therefore, the heat transfer coefficient decreases and at a concentration of 300 ppm a deterioration of heat transfer is produced. In addition, when the bubbles are detached from the liquid film, the hydrophilic groups of CTAB are adsorbed on the inner and outer surfaces of the bubbles, respectively, which increases the stability of the bubble structure. When the solution concentration is high, the molecules are closely arranged on the inner and outer surfaces of the bubble, as shown in Figure 20. The bubbles are so difficult to collapse that the heat transfer process is affected.

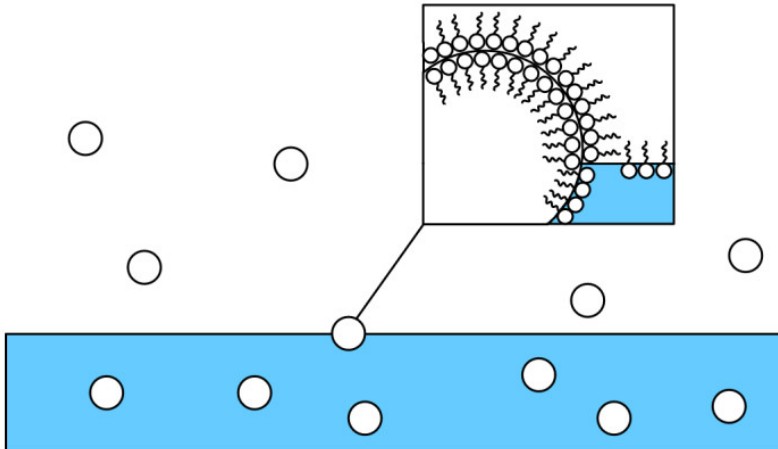

**Figure 20.** Schematic diagram of the structure of CTAB molecular adsorption bubbles.

*3.5. Effect of Ethanol–Water Mixture on the Heat Transfer Performance*

Figure 21 shows the spray cooling steady-state heat transfer coefficients at different ethanol concentrations, and Figure 22 shows the visualization of the spray chamber interior at different concentrations. As with CTAB, the surface heat transfer coefficient first increases and then decreases with increasing ethanol concentration. The maximum heat transfer coefficient of 1.79 W/cm²·K is reached at 4% ethanol volume fraction, which is 21.8% higher than that of pure water.

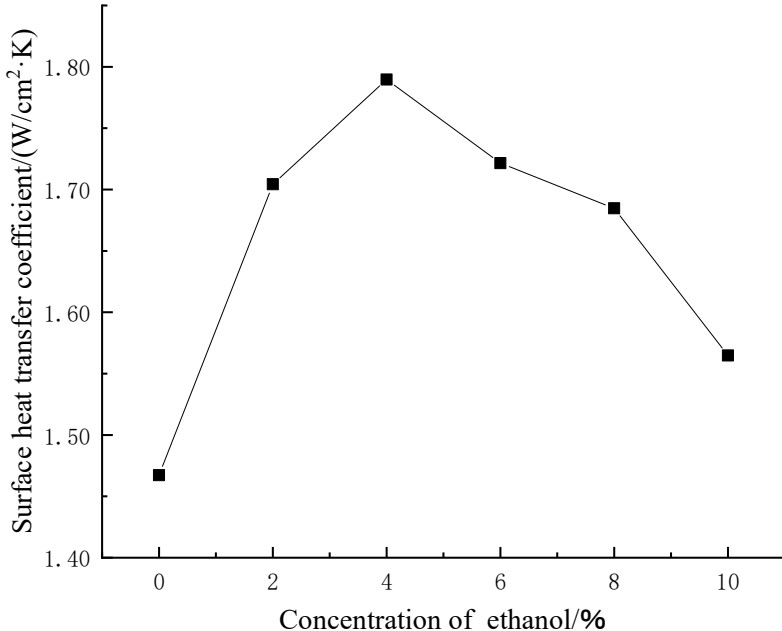

**Figure 21.** The steady-state heat transfer coefficients of spray cooling at different ethanol concentrations.

The increase in the heat transfer coefficient of the ethanol–water mixture can be attributed mainly to the evaporation of ethanol. The evaporation of ethanol at a lower surface temperature produces a vapor nucleus that acts as a secondary nucleus for the evaporation of the fluid, which is more likely to trigger the phase change of the liquid film on the heated surface and can significantly increase the heat transfer coefficient of the surface. In addition, the addition of ethanol also decreases the surface tension and contact angle of the fluid, and the droplets are more likely to break, which enhances the forced convection on the surface. The boiling point of the mixture decreases with increasing

ethanol concentration, which indicates that the two-phase regime can be initiated at a lower surface temperature, contributing to enhanced boiling heat transfer at the surface.

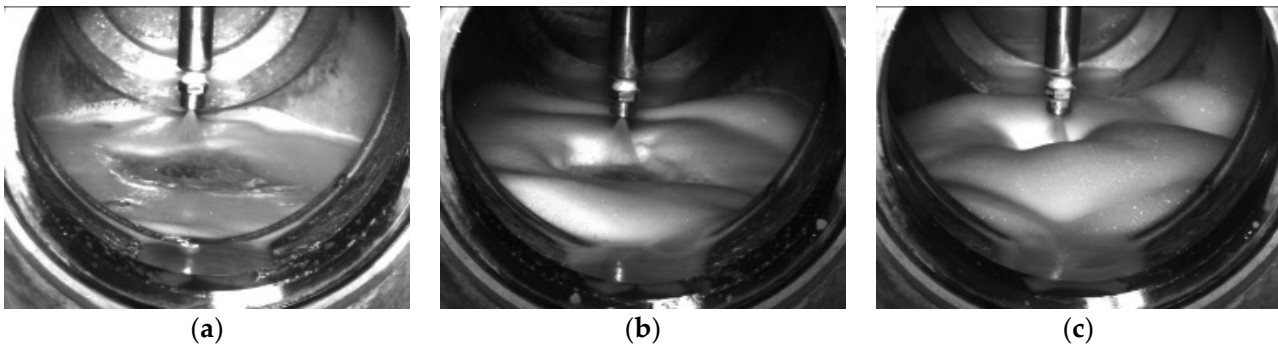

**Figure 22.** Visualization images of the foam inside the chamber at different ethanol concentrations: (**a**) 4%; (**b**) 6%; (**c**) 8%.

The specific heat and latent heat of ethanol is much lower than that of water, the heat transfer performance will be reduced if adding more ethanol in water. In addition, ethanol evaporates more easily than water, after a period ethanol gas accumulates on the surface of the liquid film, and the heat diffusion on the surface of the liquid film is blocked when the gas accumulates seriously. Ethanol aqueous solution also leads to foam generation, as shown in Figure 22, which deteriorates heat transfer at high concentrations.

### 3.6. Effect of CTAB–Ethanol–Water Mixture on the Heat Transfer Performance

Different concentrations of CTAB were added to the ethanol solution at 4% concentration. Figure 23 shows the steady-state heat transfer coefficients of the ethanol–water mixture at different CTAB concentrations, and Figure 24 shows the visualization images of the spray chamber interior at different CTAB concentrations. From Figure 23, it can be seen that initially the surface heat transfer coefficient increases and then decreases with increasing CTAB concentration. The maximum heat transfer coefficient of 1.82 W/cm$^2$·K is reached at the CTAB concentration of 200 ppm, which is 23.8% higher than that of pure water. With the concentration of 250 ppm, the heat transfer coefficient is lower than that of pure water.

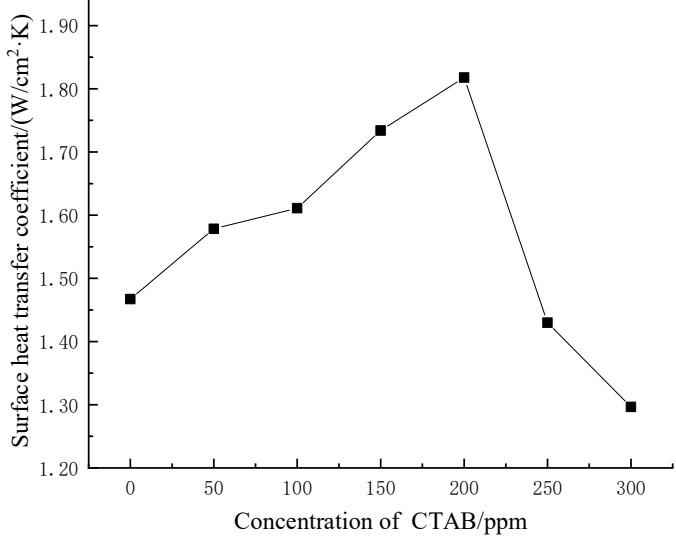

**Figure 23.** The steady-state heat transfer coefficients of the ethanol–water mixture at different CTAB concentrations.

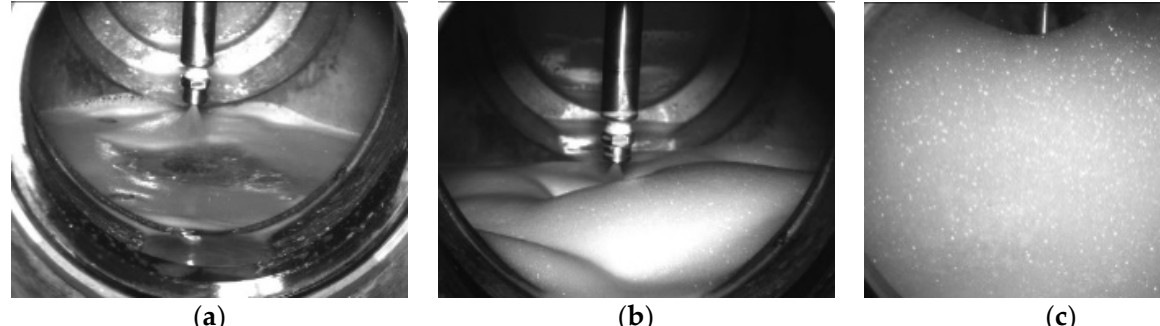

**Figure 24.** Visualization images of the foam inside the chamber of the ethanol–water mixture at different CTAB concentrations: (**a**) 20 ppm; (**b**) 40 ppm; (**c**) 60 ppm.

As seen in Figure 23, the best cooling performance of the ethanol–water mixture with CTAB is better than that of the pure water and the ethanol–water mixture but worse than that of the CTAB–water mixture. This is because compared with the ethanol–water mixture, the addition of CTAB further reduces the surface tension and contact angle of the ethanol–water mixture, while not causing more severe foaming. Due to the cavitation theory (1967), the lower the surface tension, the smaller the force required to break the vapor layer, and therefore the instability of the liquid film is enhanced, and the heat transfer performance is further enhanced. If the concentration further increases, the high foaming characteristics of ethanol and CTAB gradually dominate the heat transfer, as shown in Figure 24, resulting in the decrease in the heat transfer coefficient. However, the overall foaming effect of the ethanol–water mixture with CTAB (in Figure 24) is more severe compared to the CTAB–water mixtures (in Figure 19), which play a dominant role in affecting heat transfer.

### 3.7. Dimensionless Criterion Correlation Equations Based on Experimental Parameters

The Nusselt number, Reynolds number and Planter number are the most commonly used dimensionless numbers to characterize convective heat transfer. In this experiment, since the nozzle height and spray coverage were changed during the process, the introduction of the size coefficient h/D was considered, and the two different correlation forms were:

$$Nu = f(\text{Re}, \text{Pr}), \tag{18}$$

$$Nu = f(\text{Re}, \text{Pr}, \frac{h}{D}), \tag{19}$$

The correlation equations derived from the fitting analysis of the full experimental data were:

$$Nu = 85.49\text{Re}^{-0.673}\text{Pr}^{0.253}, \tag{20}$$

$$Nu = 18.38\text{Re}^{-0.526}\text{Pr}^{0.118}(\frac{h}{D})^{-1.39}, \tag{21}$$

Figure 25 shows a comparison between Nu calculated from the experimental parameters and Nu calculated using Equation (20). As can be seen from Figure 23, for the criterion correlation equation considering only Re and Pr, more than 90% of the experimental points fall within ±38% error of the correlation calculation results, which is suitable for some performance calculations with low accuracy requirements. The black points are the Nu obtained from each set of experimental data compared with the Nu calculated from the corresponding experimental data. However, its higher error indicates that more dimensional parameters of the spray are needed to increase the accuracy of the fitted results.

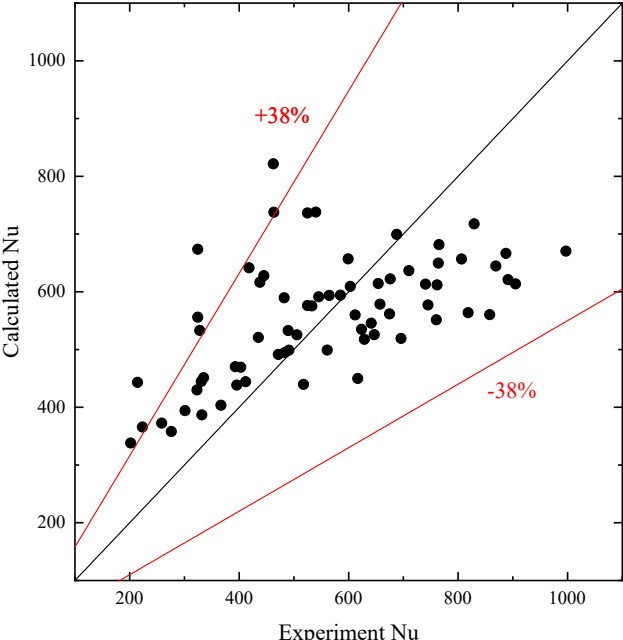

**Figure 25.** Comparison of fitted and experimental values when fitting the correlation equation using Re and Pr.

Figure 26 shows a comparison between Nu calculated from the experimental parameters and Nu calculated using Equation (21). Figure 26 shows that after considering the size parameter, more than 90% of the experimental points fall within ±25% error of the correlation calculation, indicating that the inclusion of the size parameter in the correlation equation is effective. This demonstrates that the height of the nozzle and the coverage of the spray during the spray cooling process have an effect on the calculation of the cooling performance.

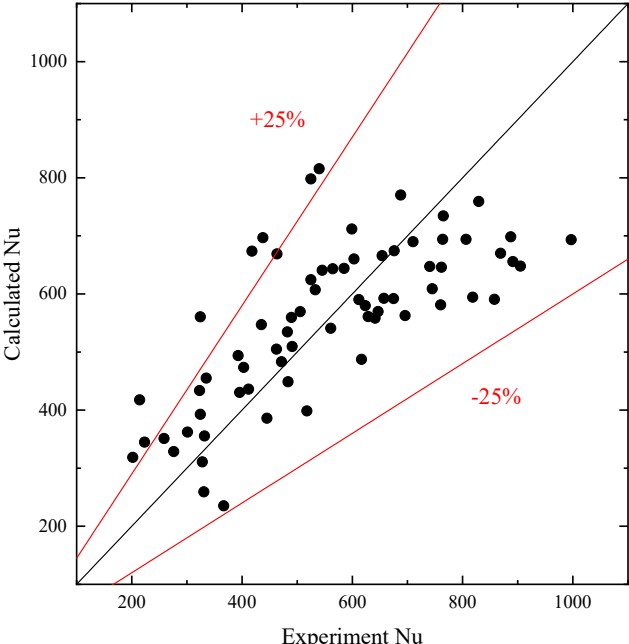

**Figure 26.** Comparison of fitted and experimental values when fitting the correlation equation using Re, Pr and h/D.

Therefore, the criterion correlation formula including Re, Pr and size coefficient can better reflect the performance of spray cooling, and the criterion correlation formula with higher accuracy is Equation (21).

*3.8. Further Application Potential for Airborne Systems*

Through the above experiments and results, the suitable medium flow rate can be selected according to the heat flux of the actual electronic equipment in the airborne spray cooling equipment, and the flow rate can be adjusted appropriately when the heat flux is small, which can save energy consumption and achieve a better cooling effect. Due to the limited space on board and the trend towards integration and miniaturization of electronic equipment, the distance between the nozzle and the heating surface is very restricted. Using the conclusion that the optimal heat transfer performance is obtained when the heating surface is not completely covered, in practical application, the distance between the nozzle and the heating surface can be appropriately reduced according to the heating capacity of different equipment, and the droplet surface coverage can be adjusted near the optimal value, which not only improves the cooling efficiency, but also saves the equipment space. In addition, without changing the system structure, a certain concentration of additives can be added appropriately, which can significantly improve the heat transfer capacity.

**4. Conclusions**

In this paper, the effects of heating heat flux, coverage of flow rate change, coverage of nozzle height change and additives on heat transfer performance were studied, and the corresponding transient heat transfer coefficients and steady-state heat transfer coefficients were calculated. Combined with the spray visualization morphology inside the cavity under several parameter variations, the change of heat transfer coefficient under several parameter variations was analyzed comprehensively, which provide reference to the question how to effectively strengthen heat transfer. The conclusions of the study are as follows:

(1) Both the transient surface heat transfer coefficient and the steady-state surface heat transfer coefficient of the simulated heating source increase as the heating heat flux increases. During the initial operation of the system under high heat flux conditions, some of the droplets will rapidly produce vapor that blocks heat transfer, resulting in a situation where the transient heat transfer coefficient increases sharply and then decreases slightly and subsequently reaches an equilibrium state.

(2) Both transient and steady-state surface heat transfer coefficients increase with increasing flow rate at high heat flux. In the case of large flow rate and small heating heat flux, the transient surface heat transfer coefficient fluctuates and the steady-state heat transfer coefficient decreases, so the heat transfer coefficient increases first and then decreases slightly with the coverage rate.

(3) When the spray coverage does not exceed the heating surface, the temperature distribution on the surface of the simulated heat source is not uniform due to the properties of the heating surface and the velocity characteristics of the spray cone. Because of the non-uniformity of the temperature distribution on the surface of the simulated heating source and the different morphology of the spray cone, optimal heat transfer performance can be achieved at lower nozzle height and smaller surface coverage.

(4) The addition of cationic surfactants (CTAB), ethanol and CTAB–ethanol mixtures all contributed to the enhancement of surface heat transfer. The addition of the CTAB–water mixture showed the greatest enhancement. The optimum enhanced heat transfer concentrations existed for all three solutions with maximum enhancement of 29.3%, 21.8% and 23.8%, respectively. However, heat transfer deterioration occurs in several solutions if the additive concentration is too high.

(5) The higher accuracy and adaptability of the dimensionless criterion correlation formula that integrates Re, Pr and size coefficient. It shows that the dimensional parame-

ters such as spray height and coverage cannot be neglected in the prediction of heat transfer ability.

(6) If it is desired to enhance heat transfer by varying the flow rate, the coverage can be increased at high heat fluxes and reduced appropriately at low and medium heat fluxes. If the heat transfer is enhanced by changing the nozzle height, the nozzle height should be reduced appropriately to reduce the coverage, encouraging the complete coverage surface in the dense area of liquid droplets. If heat transfer is enhanced by additives, the additive concentration needs to be controlled in the optimal range.

**Author Contributions:** Conceptualization, Q.N. and Y.W.; methodology, Y.W.; validation, Q.N. and N.K.; formal analysis, N.K.; investigation, Q.N.; resources, Y.W.; data curation, N.K.; writing—original draft preparation, Q.N.; writing—review and editing, Y.W.; visualization, Q.N.; supervision, Y.W.; project administration, N.K.; funding acquisition, Y.W. All authors have read and agreed to the published version of the manuscript.

**Funding:** This research was funded by the National Natural Science Foundation of China (Grant No. 51806096).

**Data Availability Statement:** The data presented in this study are openly available in [FigShare] at [https://doi.org/10.6084/m9.figshare.20502045].

**Conflicts of Interest:** The authors declare no conflict of interest.

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
