# Peer review of "The Influence of Droplet Distribution Coverage and Additives on the Heat Transfer Characteristics of Spray Cooling under the Influence of Different Parameters"

_applsci, doi:10.3390/app12189167_

Round 1
Reviewer 1 Report
This paper presents a single-nozzle open-loop spray cooling experiment platform, on which the heating surface coverage and heat transfer characteristics under different parameters were investigated. The paper is overall well written and the experiments and corresponding conclusions are sound. I recommend its publication in its present form.
Reviewer 2 Report
The paper is experimental based work and has some interesting findings relating to how visualization has been used to study the droplet heat-transfer mechanisms - mainly for 'electronic' cooling.
- The abstract starts off rather abrupty. May be the first sentence should be a bit longer.
- Few language corrections - advice to the authors to read through the paper again as spell-check may miss some errors.
- Line 155, it is stated that the heat flux is given by Eq. 4, but the heat transfer coefficient is shown.
- What's the unit for uncertainty for temperature gradient Line 161
- What sort of uncertainty are you expecting? you need to explain all the terms clearly in Eqns 5-8
Reviewer 3 Report
Please see attached file.
